# Structure, Biodegradation, and In Vitro Bioactivity of Zn–1%Mg Alloy Strengthened by High-Pressure Torsion

**DOI:** 10.3390/ma15249073

**Published:** 2022-12-19

**Authors:** Natalia Martynenko, Natalia Anisimova, Olga Rybalchenko, Mikhail Kiselevskiy, Georgy Rybalchenko, Natalia Tabachkova, Mark Zheleznyi, Diana Temralieva, Viacheslav Bazhenov, Andrey Koltygin, Andrey Sannikov, Sergey Dobatkin

**Affiliations:** 1A.A. Baikov Institute of Metallurgy and Materials Science of the Russian Academy of Sciences, Leninskiy Prospect, 49, 119334 Moscow, Russia; 2Center for Biomedical Engineering, National University of Science and Technology “MISIS”, 119049 Moscow, Russia; 3N.N. Blokhin National Medical Research Center of Oncology (N.N. Blokhin NMRCO) of the Ministry of Health of the Russian Federation, 115478 Moscow, Russia; 4P.N. Lebedev Physical Institute of the Russian Academy of Sciences, 119991 Moscow, Russia; 5A.M. Prokhorov General Physics Institute of the Russian Academy of Sciences, 119991 Moscow, Russia; 6Department of Physical Materials Science, National University of Science and Technology “MISIS”, 119049 Moscow, Russia; 7Institute of Innovative Engineering Technologies, Peoples’ Friendship University of Russia (RUDN University), 117198 Moscow, Russia; 8Casting Department, National University of Science and Technology “MISIS”, 119049 Moscow, Russia; 9Department of Metal Science and Physics of Strength, National University of Science and Technology “MISIS”, 119049 Moscow, Russia

**Keywords:** zinc alloys, high pressure torsion (HPT), microstructure, phase composition, strength, ductility, degradation, hemolysis, bioactivity

## Abstract

The effect of high-pressure torsion (HPT) on the microstructure, phase composition, mechanical characteristics, degradation rate, and bioactive properties of the Zn–1%Mg alloy is studied. An ultrafine-grained (UFG) structure with an average grain size of α-Zn equal to 890 ± 26 nm and grains and subgrains of the Mg_2_Zn_11_ and MgZn_2_ phases with a size of 50–100 nm are formed after HPT. This UFG structure leads to an increase in the ultimate tensile strength of the alloy by ~3 times with an increase in elongation to 6.3 ± 3.3% due to the formation of a basal texture. The study of corrosion resistance did not show a significant effect of HPT on the degradation rate of the alloy. In addition, no significant changes in the bioactivity of the alloy after HPT: hemolysis, cellular colonization and *Escherichia coli* growth inhibition.

## 1. Introduction

Zinc alloys are promising materials as a basis for medical products of various functionality due to their biocompatibility and biodegradability [1,2,3]. In particular, biodegradable intravascular stents [2] and scaffolds for osteosynthesis [4] have been developed on the basis of these alloys. However, insufficient strength is the main disadvantage of low-alloyed zinc alloys, which can be used for implants, metal structures, and fasteners for osteosynthesis. The first attempts to increase the strength of medical zinc alloys consisted of studying the effects of various alloying elements on their mechanical characteristics due to solid solution strengthening and the formation of strengthening phases. However, the number of metals suitable for alloying zinc-based medical alloys is limited since the possible toxicity of the alloying elements [5,6,7] should be taken into account. Therefore, alloying has limits for increasing the strength of Zn-based alloys. Previously, Li et al. showed that the addition of 0.1% of lithium to pure zinc leads to an increase in its ultimate tensile strength (UTS) from 78 ± 4.5 MPa to 146 ± 2.2 MPa. However, an increase in the Li content in the alloy up to 0.35% not only leads to its strengthening further (UTS = 137 ± 9.1 MPa) but also reduces its ductility (El = 0.3 ± 0.05%) [8]. In the case of the Zn–0.8%Mn alloy, the addition of manganese provided an increase in the UTS to 104.7 ± 2.6 MPa, and in the case of additional alloying with 0.4%Cu and 0.4%Ca, up to 117.3 ± 3.2 and 120.3 ± 6.3 MPa, respectively [9], as compared with pure Zn (UTS = ~18 MPa) [10]. Therefore, it is known that alloying can be supplemented with deformation processing in order to achieve the desired level of mechanical properties of alloys. Chen et al. showed that the use of extrusion makes it possible to increase the ultimate tensile strength of the Zn–1.5%Cu–1.5%Ag alloy from 77.2 ± 2.25 MPa in the as-cast state to 220.3 ± 1.70 MPa [11]. In addition, it is interesting to note that extrusion not only increases the strength of the alloy but also improves its ductility from 0.54 ± 0.09% to 44.13 ± 1.09% [11]. Another traditional deformation method, rolling, also makes it possible to improve the properties of medical zinc alloys. It was shown that hot rolling at 350 °C of pure zinc and Zn–4%Cu alloy leads to an increase in their strength to 151.3 MPa and 393.3 MPa, respectively, with a simultaneous increase in their ductility [12]. Additionally, the effect of severe plastic deformation (SPD) methods on the mechanical and operational properties of medical alloys based on Zn is actively studied [13]. Lui et al. showed that 12 passes of equal-channel angular pressing (ECAP) at 250 °C of the Zn–1.6%Mg alloy makes it possible to increase the strength up to 423 MPa [14]. It was possible to achieve a combination of strength and ductility equal to 474 MPa and 7%, respectively, in the case of 8 passes of ECAP at 150 °C in a rotating die [15]. Good results were also shown during investigations of the Zn–0.1%Mg alloy, where ECAP at room temperature made it possible to achieve a high level of UTS (383 MPa) in combination with excellent ductility (45.6%) [16]. Srinivasarao et al. showed the perspective of using high-pressure torsion (HPT) on pure, pre-pressed Zn, where it was possible to increase the strength from ~110 to ~150 MPa [17]. This indicates the potential of using SPD methods in order to improve the mechanical characteristics of medical zinc alloys.

The Zn–Mg system is one of the most popular alloying systems for medical zinc alloys [18]. Alloying with magnesium leads to the strengthening of pure zinc due to the formation of the Mg_2_Zn_11_ and MgZn_2_ phases. The addition of Mg does not increase the systemic toxicity of Zn–Mg alloys due to their good biocompatibility. During the implantation of a Zn-based wire for 11 months in the abdominal aorta of rats, it was noted in [19] that an increase in magnesium concentration only leads to a slight decrease in biocompatibility in vivo. In addition, a study of the bioactivity and biodegradation of Zn–Mg composites showed that the predominant corrosion of Mg-rich phases reduced the proportion of Zn^2+^ ions, which significantly improved biocompatibility. In addition, an improvement in the ability for osteogenesis and osteointegration of the composite compared with pure Zn was demonstrated after the implantation of these composites into the bone tissue of rats [20]. It was also shown in the study of in vivo biocompatibility that the implantation of Zn–0.05%Mg alloy samples in rabbits for 4, 12, and 24 weeks was not accompanied by activation of the local pro-inflammatory cellular response and mediated the formation of new bone tissue at the bone/implant interface. Degradation of the alloy in vivo does not have a toxic effect on the tissue of animal organs. In addition, it was found that the studied alloy exhibited antibacterial activity against *Escherichia coli (E. coli)* and *Staphylococcus aureus* bacteria [21]. However, our studies showed the absence of significant inhibition of *E. coli* growth after incubation with the Zn–1%Mg–0.1%Ca alloy [22]. That is, the clarification of the results of studies on the biocompatibility and bioactivity of Zn-based alloys and the replenishment of the experimental database do not lose their relevance. Thus, the purpose of this work was to study the effect of HPT on the microstructure, phase composition, mechanical characteristics, and bioactive properties of the Zn–1%Mg alloy. The study of the alloy’s bioactive properties was carried out by evaluating its effect on hemolysis, *E. coli* growth inhibition, and the ability of multipotent mesenchymal stromal cells (MMSC) to colonize the surface of alloy samples.

## 2. Materials and Methods

The Zn–1%Mg (0.98 ± 0.03 wt.%) alloy was chosen for the study in this work. The alloy was melted in an induction furnace in a clay-graphite crucible using pure Zn (99.995 wt.%) with the addition of pure Mg (99.95 wt.%). The melting process was carried out in the air without the use of a protective atmosphere and protective flux. The melt was poured into a steel mold preheated to 150 °C. The ingot obtained as a result of casting was annealed for 20 h at 340 °C. The ingot was quenched in water to fix the high-temperature state. Samples with a diameter of 20 mm and a thickness of 1.5 mm were cut from the obtained ingot for the HPT process. The deformation was carried out under a pressure of 4 GPa at room temperature. The total deformation degree of samples after HPT was 5.7 (number of turns N = 10). The total deformation degree was calculated according to the method described in [23]. A schematic representation of the Bridgman anvil [23] used for HPT and a scheme for cutting a sample for research are shown in Figure 1.

The alloy’s microstructure in the initial state was studied using the scanning electron microscopy (SEM) method. For this, a JSM-7001F (JEOL; Tokyo, Japan) scanning electron microscope we used. The study of the alloy’s microstructure after HPT was carried out at half of the sample radius. The microstructure was studied by transmission electron microscopy (TEM). A JEM-2100 (JEOL; Tokyo, Japan) transmission electron microscope operating at 200 kV and equipped with an energy-dispersive spectrometer (EDS) was used to study the deformed microstructure. The samples for the study were prepared by manual mechanical thinning of up to 0.12 mm. Further preparation was carried out by the electrochemical method on a TenuPol-5 Struers unit (Struers, Copenhagen, Denmark). The polishing electrolyte consisted of a mixture of perchloric acid, ethanol, butoxyethanol, and distilled water. The electropolishing process was carried out at a voltage of 20 V. The phase composition was studied using X-ray diffraction (XRD) on a Bruker D8 Advance diffractometer (CuK_α_ radiation, λ = 1.54 Å; Karlsruhe, Germany). Further processing of the results was carried out using the Rietveld method with the use of a Bruker DIFFRAC.EVA™ (Karlsruhe, Germany), DIFFRAC.TOPAS™ (Karlsruhe, Germany) and ICDD PDF-2 2020™ (Newtown Square, PA, USA) software. The microhardness was measured using a 402 MVD Instron Wolpert Wilson Instruments microhardness tester under a load of 50 g and an indentation time of 10 s. The mechanical properties were evaluated using flat samples with a length of 5.75 mm and a cross-section of 2 × 1 mm. The study was conducted on an Instron 3382 testing machine at room temperature.

The study of the corrosion resistance of the alloy was carried out by the immersion method at a temperature of T = 37 °C. For this, samples were used in the form of a 1/8 disk with a diameter of 20 mm. The tests were performed in a complete growth medium based on Dulbecco’s Modified Eagle Medium (DMEM; pH = 7), supplemented with 10% Fetal Bovine Serum (FBS), 1% penicillin/streptomycin, and 4 mM L-glutamine (all PanEco, Moscow, Russia), mimicking properties of the intercellular environment in the patient’s body. The alloy samples were immersed in 70% ethanol for 2 h to sterilize them before tests. Pre-weighed on a Sartorius M2P Micro Balances Pro 11 (certified by ISO 9001) (Data Weighing Systems, Inc, Wood Dale, USA; three digits per mg) samples (N = 3) were immersed in DMEM for 8 days. The samples were thoroughly dried after immersion tests. The study of the composition of degradation products was carried out using a JSM-7001F (JEOL; Tokyo, Japan) scanning electron microscope. The samples were cleaned in a mixture of 100 g ammonium persulfate ((NH_4_)_2_S_2_O_8_) and reagent water (to make 1000 mL) for 5 min to remove degradation products [24], and then samples were thoroughly air-dried again under sterile conditions and weighed. The degradation rate (DR, mm/year) of the Zn–1%Mg alloy was calculated according to the equation [24]:(1)$$\mathrm{DR}=8.76\;\times \;10^{4}\;\times \;\frac{\Delta m}{A\times t\times \rho }$$
where Δ*m* is the mass loss in grams, *t* is the immersion time in hours, *A* is the specimen surface area in cm^2^, and *ρ* is the density of the alloy in g/cm^3^.

The in vitro study of bioactive properties was carried out on samples in the form of 1/8 discs with a diameter of 20 mm (at least three samples per each type of study). All samples were sterilized before testing, as described above. Bioactive properties were assessed by studying the hemolysis and lactate dehydrogenase (LDH) activity of MMSCs that colonized the surface of samples. The methodology of the conducted tests and statistical analysis of the results were described in detail earlier in [22]. In particular, to assess hemolytic activity, 2 mL of C57 BL/6 mouse blood mixed with phosphate-buffered saline (PBS) (PanEco, Moscow, Russia) in a 1:5 ratio were incubated with alloy samples in a 24-well plate (Nunc, Waltham, MA, USA) for 4 h at 37 °C. In control, the blood suspension was incubated without alloys under similar conditions. Then, hemoglobin absorption in the supernatant was measured using a Spark plate reader (Tecan, Switzerland) at 540 nm versus 690 nm. The ratio of absorption in the wells with alloy samples to the control was calculated to assess the hemolytic activity of the samples (the value was expressed as a percentage).

The effect of alloys on cellular colonization was studied in the LDH test. MMSCs from the N.N. Blokhin NMRCO collection were used as a cell model. MMSCs were cultivated as described earlier [25]. Before the experiment, the cells were treated with trypsin in the logarithmic growth. After washing by DMEM, the cells were suspended in a DMEM-based complete growth medium at a concentration of 840,000 cells/mL. Tested alloy samples were placed in the wells of the 24-wells plate. Precisely 20 μL of the MMSCs in the complete growth medium was seeded on alloy samples, preincubated for 30 min at 37 °C in an atmosphere with 5% carbon dioxide, and cultivated after the addition of 2 mL of the complete growth medium for 10 days under the same conditions. The MMSCs seeded on the bottom of the empty wells were used as a control. The medium was changed every 2 days. The LDH activity was studied using Pierce LDH Cytotoxicity Assay Kits (Thermo Scientific, Waltham, MA, USA) in accordance with the manufacturer’s method by measuring adsorption at 450 nm against 620 nm (A450–A620) with the plate reader. The cellular colonization was studied with an automated digital microscope (Lionheart LX, BioTek, Santa Clara, CA, USA) after staining cells with Calcein AM (Sigma-Aldrich, USA).

An 18 h *E. coli* culture (collection of the N.N. Blokhin NMRCO) in Mueller–Hinton broth (Pronadisa, Torrejon de Ardoz, Spain) was used to study the bacterial growth inhibition under the influence of the alloys. Exactly 20 μL of the *E. coli* culture was seeded on alloy samples, preincubated for 30 min at 37 °C in an atmosphere with 5% carbon dioxide, and cultivated after the addition of 2 mL of Mueller–Hinton broth for 1 day under the same conditions. For the control, the bacteria were incubated without samples of alloys under the same conditions. In order to evaluate the growth of *E. coli* bacteria, Alamar blue (Invitrogen, Waltham, MA, USA) was used in accordance with the manufacturer’s instructions. During the experiment, the fluorescence Ex570/Em620 was measured with the plate reader. The study results were presented as the percentage of fluorescence of live bacteria in wells with alloy to the control. In addition, a Live/Dead BacLight Bacterial Viability Kit (Invitrogen, USA) for bacteria staining was used. The result was recorded using the Lionheart FX automated digital microscope.

The animal procedures and experiments with cells were assessed and approved by the Ethical Committee of the N.N. Blokhin NMRCO (protocol #AAAAA-A19-119061190077-2, approval date: 12 May 2021).

The results of statistical analysis were presented as a mean ± standard deviation (Mean ± SD). Comparisons between the two groups were made using the Student’s test. Differences were considered statistically significant at *p* < 0.05.

## 3. Results

Figure 2 shows the results of a study on the microstructure of the Zn–1%Mg alloy before and after HPT.

The alloy’s microstructure in the annealed state consists of α-Zn dendritic cells with an average size of ~35 µm, surrounded by an interlayer of a phase mixture rich in magnesium (Figure 1a). Significant refinement of the alloy’s microstructure (both α-Zn dendrites and the phase mixture) occurs after HPT. At the same time, the EDS analysis showed that the equiaxed grains (point 1) consist of pure Zn, while the phase formed at the edges (points 2 and 3) is enriched in Mg (Figure 2b–d). The average grain size of α-Zn after HPT is 890 ± 26 nm (Figure 1b), while the Mg-rich phase consists of grains and subgrains with a size of 50–100 nm (Figure 2c). Moreover, the formation of finely dispersed particles ~100 nm in size is observed in the alloy structure in addition to the grain-boundary phase.

The results of the study of the alloy’s phase composition are shown in Figure 3. The studies have shown that the alloy in both states consists of three types of phases: α-Zn (P6_3_/mmc), Mg_2_Zn_11_ phase (Pm$\bar{3}$), and MgZn_2_ phase (P6_3_/mmc). It should be noted that HPT does not lead to a significant increase in the mass fraction of the Mg_2_Zn_11_ and MgZn_2_ phases; however, there is a tendency to increase it within the error. The mass fraction of the Mg_2_Zn_11_ phase is 7.3 ± 1.6 and 14.2 ± 6.6%, and the mass fraction of the MgZn_2_ phase is 3.0 ± 0.8 and 5.7 ± 2.9% for the annealed and HPT-treated alloy, respectively (Table 1). Additionally, it should be noted that a significant increase in the intensity of the (00.2)_Zn_ and (00.4)_Zn_ lines with a weakening of other lines is discovered after HPT. This indicates the formation of a strong basal texture in the alloy after HPT.

Figure 4 shows the results of a study of the microhardness of the Zn–1%Mg alloy before and after HPT.

The microhardness of the alloy in the initial state is 801 ± 40 MPa. The microhardness of the alloy after HPT is distributed unevenly over the sample section. The microhardness values in the center of the sample are significantly lower than at its edges. This is a typical situation for HPT-treated materials since the value of the deformation degree after HPT is directly proportional to the sample radius [23]. However, the inhomogeneity of the microstructure (and hence the inhomogeneity of microhardness) decreases with an increase in the number of revolutions. In our case, the alloy’s microhardness becomes uniform at a distance of more than 2 mm from the sample center. Therefore, the studies of the mechanical properties of the alloy after HPT were carried out in the middle of the sample radius. The microhardness of the Zn–1%Mg alloy after HPT, measured at the middle of the sample radius, is 1132 ± 29 MPa.

Table 2 presents the results of a study of the mechanical characteristics of the Zn–1%Mg alloy before and after HPT. The yield strength of the alloy in the initial state is 153 ± 7 MPa, while the ultimate tensile strength is 156 ± 3 MPa. The ductility of the annealed alloy was almost at the zero level and amounted to 0.2 ± 0.04%. The yield strength of the alloy after HPT increases almost 2.5 times (up to 374 ± 5 MPa), and the ultimate tensile strength increases almost 3 times (up to 459 ± 25 MPa). It is also interesting to note that the ductility of the alloy after HPT increases to 6.3 ± 3.3%.

The results of the study of the alloy’s corrosion resistance and elemental SEM-EDS-mapping of degradation products are presented in Figure 5. The studies have shown that the structure caused by HPT does not lead to a deterioration in the corrosion resistance of the alloy (Figure 5a). The degradation rate of the annealed alloy is 0.16 ± 0.03 mm/y. The DR value of the alloy after HPT is 0.21 ± 0.03 mm/y. At the same time, it should be noted that the degradation of the alloy, both in the annealed and in the HPT-treated states, occurs mainly in places rich in magnesium (Figure 5b,c), probably because magnesium-rich Mg_2_Zn_11_ and MgZn_2_ phases have a higher degradation rate than pure zinc. Additionally, it should be noted that the composition of the degradation products has identical composition for the alloy in both states. O, P, Cl and Ca, which characterize the composition of the test medium, were also found in the degradation products in addition to Zn and Mg, which are part of the alloy [26].

The estimation of the level of induced hemolysis was conducted to evaluate the in vitro biocompatibility of the alloy before and after HPT (Table 3). The studies have shown that the effect of the alloy in both states does not lead to a significant change in the level of hemolysis compared with the control (*p* > 0.05). These results also indicate that HPT does not lead to an increase in the hemolytic activity of the Zn–1%Mg alloy in comparison with the initial state.

The study of the effect of the alloy on bacterial growth was carried out using an *E. coli* culture, which was incubated on the surface of alloy samples for 1 day. The study of fluorescence microscopy data after differential staining of the samples showed that the colony formation of bacterial culture on the surface of alloys of both types was intense, with a minimum number of dead cells (Figure 6). This result was confirmed by quantitative analysis data, allowing assessment of the viability of the bacteria after 24 h of incubation together with alloys (Table 3). It was found that the Zn–1%Mg alloy, both in the annealed state and after HPT, does not demonstrate pronounced antibacterial properties, regardless of the treatment used, since it was not possible to identify a significant decrease in the viability of the bacterial culture at the end of the incubation period.

Cellular colonization of the surface of Zn–1%Mg alloy samples before and after HPT is shown in Figure 7.

The studies of the activity of the cells with osteogenic potential showed that the effect of the alloy in both states did not mediate the cytotoxic effect, accompanied by the inhibition of LDH activity of MMSC in comparison with the control (*p* > 0.05), and did not inhibit cell colonization of the surface of the samples. It should also be noted that the absence of a significant difference between the value of the studied parameter for the alloy in the annealed state and after HPT was observed (*p* > 0.05). However, analysis of the fluorescence microscopy data of the surface of the alloys showed that, although the cells were alive and their membranes did not have ruptures, their morphology was different from the cells in the control. In particular, MMSCs in the control had an elongated shape, a rather large size, and high confluence due to spreading at the bottom of the plate well, occupying almost the entire area of the well. On the contrary, the cells on the surface of the alloys had a spherical shape, a smaller size, and very weakly spread over the area of the sample.

## 4. Discussion

In this work, the effect of HPT on the structure and properties of a bioresorbable Zn–1%Mg alloy as potential material for medical use was studied. Special attention was paid to aspects of microstructure, phase composition, mechanical properties, and corrosion resistance. In addition, the dependence of in vitro bioactivity on alloy processing was evaluated. The conducted studies have shown that HPT significantly improves the mechanical properties of the Zn–1%Mg alloy. Therefore, the yield stress of the alloy after HPT increased by ~2.5 times and the ultimate tensile strength by ~3 times. This increase in strength is due to the formation of an ultrafine-grained structure in the alloy after HPT. Previously, it was shown in [27] that grain refinement after HPT in the Zn–0.8% Ag alloy provided a significant increase in its microhardness. It is also interesting to note that HPT leads not only to an improvement in the strength properties of the alloy but also significantly increases its ductility. The increase in ductility is probably associated with the formation of a favorable type of alloy texture. It was shown in [28] that grain refinement and the formation of a favorable texture in the Zn–1%Cu alloy after ECAP lead not only to an increase in its strength but also an increase the ductility from 7.3 to 94.2%. In our case, the increase in the basal lines (00.2) and (00.4) on the X-ray diffraction pattern of the alloy after HPT indicates the formation of an intense basal texture after deformation (Figure 3b). It is known that the formation of a basal texture leads to an improvement in ductility in metals with a hexagonal close-packed lattice [29,30]. At the same time, the increase in ductility significantly expands the possibilities of using the developed alloy in orthopedics, making it possible to manufacture products of various shapes and purposes.

Table 4 compares the mechanical characteristics measured in this work with the data obtained earlier for alloys of the Zn–Mg system. As can be seen from the data presented in the table, the obtained strength values are significantly higher than those for alloys of the Zn–Mg system. Thus, the use of extrusion makes it possible to increase the strength of Zn–Mg alloys up to 342–360 MPa [31,32,33]. The use of rolling results in a strength value not exceeding 270 MPa [34,35]. It should be noted that the values of relative elongation in all cases have an acceptable level for the considered application. A strength level close to that obtained in this work (UTS = 423 MPa) was achieved by deformation with another SPD method—ECAP at 150 °C. However, the content of Mg, and, consequently, the mass fraction of the Mg_2_Zn_11_ and MgZn_2_ phases, is higher in the ECAP-treated alloy [14].

A study of the corrosion resistance of the Zn–1%Mg alloy showed that the UFG structure formed after HPT did not lead to a significant change in the alloy degradation rate (which is 0.16 ± 0.03 and 0.21 ± 0.03 mm/y for the annealed and HPT-treated states, respectively). The probable reason for this behavior is the unchanged phase composition of the alloy after HPT. The structure of the HPT-treated alloy consists of three phases (α-Zn, Mg_2_Zn_11_, and MgZn_2_) that are the same as in the case of the annealed alloy. The mass fraction and arrangement of magnesium-rich phases (along the α-Zn boundaries) also do not change. Similar data were previously obtained on alloys of the Zn–Mg–Zr system, where deformation by hot rolling also did not lead to a significant increase in the degradation rate [36].

It is logical that with the indicated absence of a change in the degradation rate and phase composition, the signs of a significant change in various aspects of the biological activity of the Zn–1% Mg alloy after HPT were not observed. Thus, the biocompatibility of the studied alloy, both in the initial state and after HPT, was confirmed. This conclusion was made because of no increase in the level of hemolysis and inhibition of cellular LDH activity after incubation with the studied samples. In addition, our studies have shown that the alloy in both states does not inhibit the colonization of the surface of the samples of the alloy with MMSCs osteogenic potential. Therefore, no cytotoxic effect of the studied alloy was revealed, which makes it possible to attribute it (regardless of processing) to biocompatible materials. It is interesting that a significant change in cell morphology during incubation on the surface of the alloys was detected against the background of the preservation of the cellular activity of MMSC by fluorescence microscopy. It is likely that it is associated with the destruction of the cell substrate due to the constant degradation of the surface of the alloy samples but not with the cytopathogenic effect of biodegradation products. This destruction prevents full-fledged cell adhesion and the formation of close intercellular interactions. Indirectly, this conclusion is confirmed by the fact that it was not possible to observe the presence of bactericidal activity in the samples of the studied alloy during the research. At the same time, there is reason to believe that the degradation products of the Zn–1%Mg alloy could stimulate colonization of MMSCs of the surface of alloy samples due to bioactive elements of its composition. This effect favorably distinguishes the Zn–1%Mg alloy from other biodegradable materials that are promising for use in medicine [37,38,39]. Thus, the localization of the degradation process was observed along the grain boundaries of α-Zn, that is, in the places of localization of phases rich in Mg, in Zn–0.5%Mg and Zn–1%Mg alloys [35]. Based on this, it can be concluded that the degradation products of the Zn–1%Mg alloy are rich in Mg^2+^ ions, which can stimulate the process of surface colonization of samples. This conclusion is confirmed by a study [40], which showed that the release of Mg^2+^ ions during the degradation of biodegradable membranes based on polylactic acid (PLA) -containing Mg improves MMSCs proliferation and enhances osteoinductive capacity. A similar effect was demonstrated in [41], where it was shown that extracts of pure Mg and Mg–3%Zn and Mg–2%Zn–1%Mn alloys containing a high concentration of Mg^2+^ ions have a significant stimulating effect on osteogenic differentiation.

Based on the foregoing, HPT is a promising method for obtaining biodegradable metal structures based on the Zn–1%Mg alloy. The rationale for this conclusion is the fact that after HPT by 3 times, the strengthening of the alloy with a simultaneous increase in ductility was observed. At the same time, the corrosion resistance and bioactive properties of the alloy remain almost unchanged after HPT. Thus, the obtained results indicate the prospects for the development of submersible orthopedic biodegradable metal structures and implants based on the Zn–1%Mg alloy after HPT.

## 5. Conclusions

The α-Zn grain size after HPT refines to 890 ± 26 nm, while the Mg-rich phase consists of grains, subgrains, and particles 50–100 nm in size.The alloy in both states consists of pure α-Zn, as well as Mg_2_Zn_11_ and MgZn_2_ phases. The XRD method also revealed the formation of an intense basal texture in the alloy after HPT.The microstructure caused by HPT does not lead to a decrease in the resistance of the Zn–1%Mg alloy to chemical corrosion. The value of the degradation rate measured by the immersion method is 0.16 ± 0.03 mm/y and 0.21 ± 0.03 mm/y for the annealed and HPT-treated alloy, respectively.The formation of the UFG structure increases the YS of the alloy by ~2.5 times and the UTS by ~3 times. HPT increased the microhardness of the alloy from 801 ± 40 MPa to 1132 ± 29 MPa. In addition, an increase in ductility of the alloy to 6.3 ± 3.3% is observed due to the formation of a basal texture.HPT does not lead to a change of the hemolysis level and colonization of Zn–1%Mg alloy surface by cells with osteogenic potentialThe Zn–1%Mg alloy, both in the initial state and after HPT, does not inhibit the growth of *E. coli* bacteria during co-incubation

## Figures and Tables

**Figure 1 materials-15-09073-f001:**
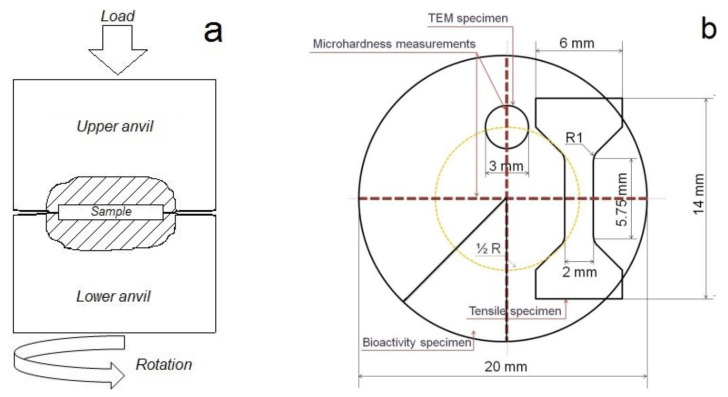
Schematics of the HPT process (**a**) and representation showing the locations for microhardness measurements on the HPT disc and the cutting position of TEM; bioactivity and tensile samples inside it (**b**).

**Figure 2 materials-15-09073-f002:**
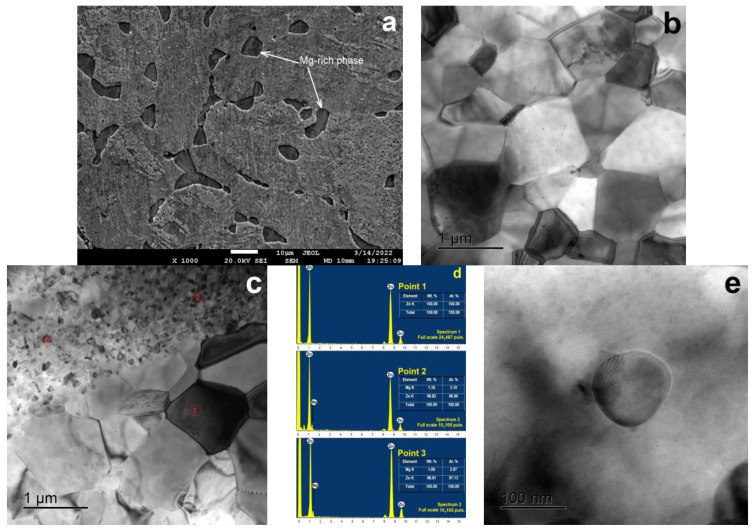
SEM-image (SE-contrast) structure of the Zn–1%Mg alloy in the annealed state (**a**) and the TEM-images (bright field) of the structure of the Zn–1%Mg alloy after HPT (**b**,**c**); TEM-EDS point analysis result of the Zn–Ca particle (**d**) and HRTEM-image of intermetallic particles based on Zn–Mg (**e**).

**Figure 3 materials-15-09073-f003:**
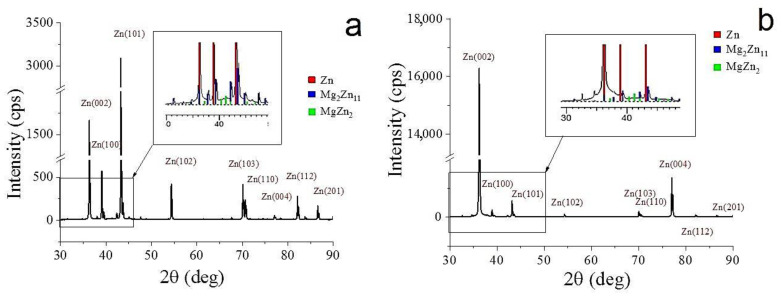
X-ray diffraction patterns of the Zn–1%Mg alloy before (**a**) and after HPT (**b**).

**Figure 4 materials-15-09073-f004:**
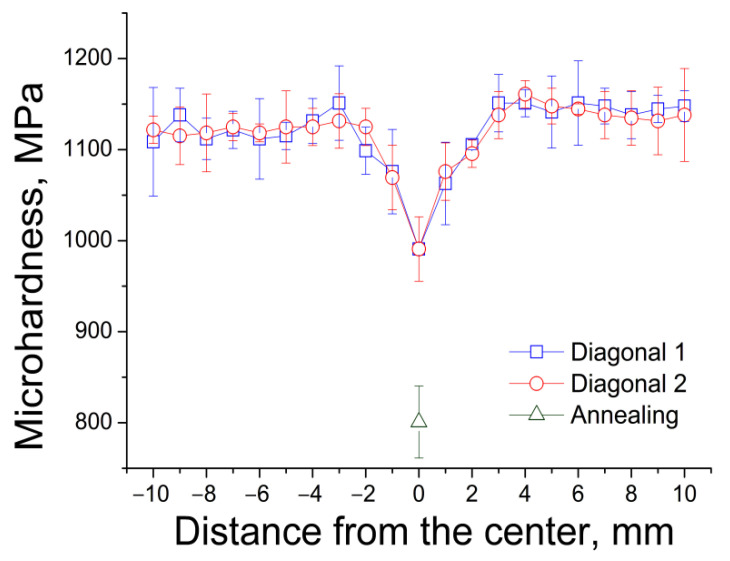
Distribution of microhardness of the alloy over the diameter of the sample after HPT.

**Figure 5 materials-15-09073-f005:**
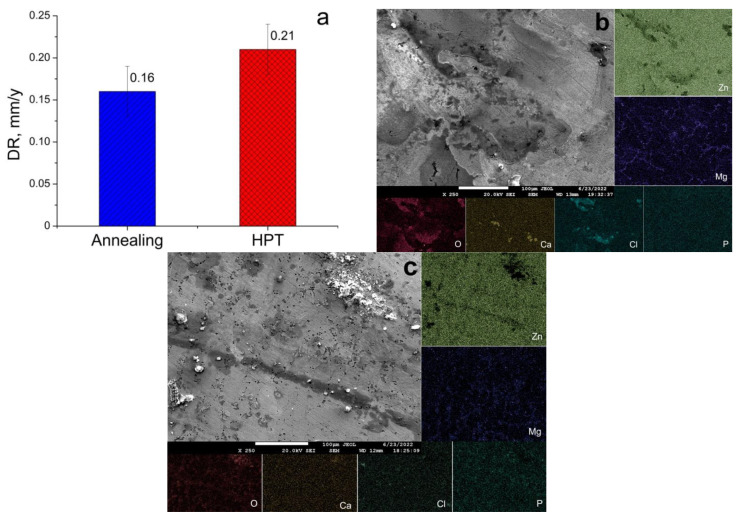
Degradation rate (DR) of the alloy in the annealed and HPT-treated conditions in vitro (**a**) and SEM images of corroded surfaces and elemental mapping of annealed (**b**) and HPT-treated (**c**) alloy samples after incubation in DMEM for 8 days.

**Figure 6 materials-15-09073-f006:**
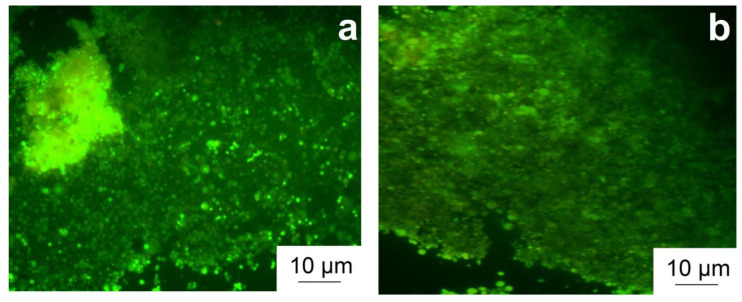
Growth of *E. coli* bacteria on the surface of samples of the annealed (**a**) and HPT-treated (**b**) alloy. Cells with a compromised membrane that are considered to be dead or dying stain red, whereas live cells with an intact membrane stain green.

**Figure 7 materials-15-09073-f007:**
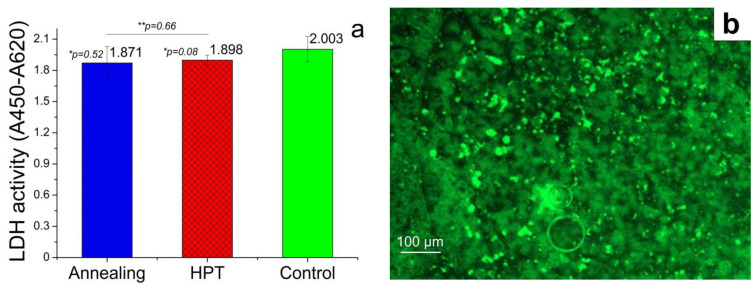
Multipotent mesenchymal stromal cell lactate dehydrogenase activity (**a**) and colonization of the surface of samples the annealed (**b**) and HPT-treated (**c**) alloy in comparison with cellular control (**d**). Live MMSC with an intact membrane stain green after treatment by Calcein AM.

**Table 1 materials-15-09073-t001:** Results of XRD analysis (Rietveld method) of the Zn–1%Mg alloy in the initial state and after HPT processing.

State	Phase	Space Group	Mass Fraction, wt. %	a (Å)	c (Å)
Annealing	α-Zn	P6_3_/mmc (194)	89.7 ± 2.2	2.666 ± 0.001	4.951 ± 0.001
	Mg_2_Zn_11_	$\mathrm{Pm}\bar{3}$ (200)	7.3 ± 1.6	8.544 ± 0.001	8.544 ± 0.001
	MgZn_2_	P6_3_/mmc (194)	3.0 ± 0.8	5.210 ± 0.003	8.772 ± 0.009
HTP	α-Zn	P6_3_/mmc (194)	80.1 ± 9.2	2.666 ± 0.001	4.949 ± 0.001
	Mg_2_Zn_11_	$\mathrm{Pm}\bar{3}$ (200)	14.2 ± 6.6	8.561 ± 0.004	8.561 ± 0.004
	MgZn_2_	P6_3_/mmc (194)	5.7 ± 2.9	4.997 ± 0.009	8.889 ± 0.016

**Table 2 materials-15-09073-t002:** The mechanical characteristics of the Zn–1%Mg alloy in the annealed and HPT-treated states (YS—yield stress; UTS—ultimate tensile strength; El—elongation; HV—microhardness).

State	YS, MPa	UTS, MPa	El, %	HV, MPa
Annealing	153 ± 7	156 ± 3	0.2 ± 0.04	801 ± 40
HPT	374 ± 5	459 ± 25	6.3 ± 3.3	1132 ± 29

**Table 3 materials-15-09073-t003:** Hemolytic activity of RBCs and growth of *E. coli* with samples of the annealed and HPT-treated alloy in comparison with control (SD—standard deviation; * the difference from the control; ** the difference from the alloy after HPT treatment).

Parameter	State of alloy	Mean, %	SD, %	* *p*	** *p*
Hemolysis	Annealing	98	6	0.078	-
	HPT	104	3	0.061	0.053
	Control	100	1	-	-
Growth of *E. coli* bacteria	Annealing	94	10	0.76	-
	HPT	88	12	0.69	0.28

**Table 4 materials-15-09073-t004:** Mechanical properties of bioresorbable Zn–Mg alloys after different deformation treatments.

Alloy	Treatment	YS, MPa	UTS, MPa	El, %	Ref.
Zn–1%Mg	HPT	374 ± 5	459 ± 25	6.3 ± 3.3	Present work
Zn–1%Mg	Conventional extrusion at 200 °C	289	320	32	[31]
Zn–1%Mg	Extrusion at 250 °C	180 ± 7.3	340 ± 15.6	6 ± 1.1	[32]
Zn–1.2%Mg	Extrusion at 250 °C	219.61 ± 15.42	362.64 ± 4.87	21.31 ± 2.31	[33]
Zn–0.8%Mg	Multiple rolling at 250 °C	260	268	7.2	[34]
Zn–1%Mg	Cold rolling	222 ± 6	260 ± 8	11 ± 1	[35]
Zn–1.6%Mg	ECAP at 150 °C ECAP, 12 passes	361	423	5.2	[14]

## Data Availability

All the data required to reproduce these experiments are present in the article.

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
