# Peer review of "Structure, Biodegradation, and In Vitro Bioactivity of Zn–1%Mg Alloy Strengthened by High-Pressure Torsion"

_materials, 2022, doi:10.3390/ma15249073_

Round 1
Reviewer 1 Report
In this paper, the authors have studied the influence of severe plastic deformation SPD using HPT on the microstructure, phase composition, mechanical characteristics, degradation rate, and bioactive properties of the Zn-1%Mg alloy. The HPT processing contributes to the refinement of the grain and phase and so the mechanical properties of the UTS and elongation. On the other hand, the corrosion resistance and the bioactivity did not change after the HPT processing. The results of the paper were well presented and discussed. It is exciting research; the reviewer suggests accepting this paper for publication in the materials after a minor revision to cover the following comments.
1- What is the author means by (The total deformation degree of samples after HPT was 5.7)? Did they mean the imposed stain? If so, please insert the equation used for this calculation and give the value of the parameters used in the current study.
2- Why the HPT processing parameters were limited to a pressure of 4 GPa and 10 revolutions? Is that based on a previous study, so a reference must be mentioned?
3- Can authors indicate the Mg-rich phase grains in figure 2 by arrows?
4- Please add any information about the microhardness measurement in the materials and methods section and conclusions.
Sincerely yours,
Author Response
Dear Editor,
We would like to thank the reviewers for their valuable comments. We have considered them carefully and made some changes to the manuscript to address the criticisms. The alterations made are highlighted in colour. Our response to the critical points made by the reviewers is provided below.
Reviewer #1:
Q1: What is the author means by (The total deformation degree of samples after HPT was 5.7)? Did they mean the imposed stain? If so, please insert the equation used for this calculation and give the value of the parameters used in the current study.
Response to Q1: The total deformation degree was calculated according to the method described in [Zhilyaev, A.P.; Langdon, T.G. Using high-pressure torsion for metal processing: Fundamentals and applications. Prog Mater Sci. 2008, 53, 893–979. https://doi.org/10.1016/j.pmatsci.2008.03.002].
Q2: Why the HPT processing parameters were limited to a pressure of 4 GPa and 10 revolutions? Is that based on a previous study, so a reference must be mentioned?
Response to Q2: The choice of pressure was carried out experimentally. We used three variations: 2, 4 and 6 GPa. The strength of the alloy deformed under a pressure of 4 GPa was higher compared to the alloy deformed under a pressure of 2 GPa. When the pressure was increased up to 6 GPa, the samples were cracked.
The number of revolutions is due to the desire to obtain a more uniform microstructure. It is known that an increase in the number of revolutions during HPT makes it possible to obtain more uniform microstructure [Zhilyaev, A.P.; Langdon, T.G. Using high-pressure torsion for metal processing: Fundamentals and applications. Prog Mater Sci. 2008, 53, 893–979. https://doi.org/10.1016/j.pmatsci.2008.03.002]. However, increasing the number of revolutions above 10 leads to cracking of samples.
Q3: Can authors indicate the Mg-rich phase grains in figure 2 by arrows?
Response to Q3: Figure 2 has been corrected.
Q4: Please add any information about the microhardness measurement in the materials and methods section and conclusions.
Response to Q4: A sentence has been added to section 2: «The microhardness was measured using on a 402 MVD Instron Wolpert Wilson Instruments microhardness tester under a load of 50 g and an indentation time of 10 s». The 4st conclusion has been added: « HPT increased the microhardness of the alloy from 801 ± 40 MPa to 1132 ± 29 MPa».
Reviewer #2:
Q1: Line 75, This indicates the potential of using SPD methods: Here, the text just talks about HPT in the last sentence. Should the SPD be HPT? Or HPT is part of SPD methods?
Response to Q1: We believe that the study of the effect of various SPD methods on the structure and properties of Zn-based alloys deserves attention. In this study, we chose the HPT method. However, we are currently investigating other SPD methods, for example equal-channel angular pressing. In the future, we hope to conduct a comparative analysis of the methods.
Q2: Line 112 and 113, after HPT was 5.7: What is unit of 5.7 and definition of total deformation degree?
Response to Q2: It means the equivalent von Mises strain described in [Zhilyaev, A.P.; Langdon, T.G. Using high-pressure torsion for metal processing: Fundamentals and applications. Prog Mater Sci. 2008, 53, 893–979. https://doi.org/10.1016/j.pmatsci.2008.03.002]. The total deformation degree is a dimensionless quantity
Q3: Line 118-120: This sentence contains two verbs.
Response to Q3: The sentence has been revised: «The study of the microstructure of the alloy after HPT was carried out at the half of the sample radius. The microstructure was studied by transmission electron microscopy (TEM)».
Q4: Line 135: What does the 1/8 disk mean?
Response to Q4: The disc after HPT was cut into 8 equal parts for biocompatibility studies. The details were shown in Figure 1b.
Q5: In Fig 2c: The label of 1, 2 and 3 is too small to see. Either change the color or size will do.
Response to Q5: Figure 2 c has been corrected.
Q6: There are many places in the sentences, a comma should be put to make the sentence more readable. For example, line 81, line 117, line 171, line 190.
Response to Q6: The sentences have been revised.
Q7: MMSC already defined in line 102, should not be defined again at line 159.
Response to Q7: The sentence has been revised.
Further minor changes (marked in the revised manuscript) have also been made. We now present the revised paper to the judgement of the reviewers and yourself as the Editor.
Sincerely,
on behalf of all authors
Prof. Sergey Dobatkin, Dr. of Sci.,
Head of Laboratory of
Physical Metallurgy of Non-Ferrous
Metals, A.A. Baikov Institute of
Metallurgy and Materials Science,
Russian Academy of Sciences,
Moscow, Russia
Phone: +7 499 135 7743
sdobatkin@imet.ac.ru

Reviewer 2 Report
In this work, the effect of Zn-1% Mg was strengthened by high pressure torsion(HPT) treatment. The microstructure, phase composition, mechanical properties, corrosion resistance and the dependence of bioactivity in vitro on alloy processing were evaluated. The conducted studies have shown that HPT significantly improves the mechanical properties of the Zn-1%Mg alloy, while does not affect the bioactivity too much. This is a new piece of material process work with substantial progress for the bio-applications and should be published on Materials. However, more improvements can be done, to make it easier to read:
Line 75, This indicates the potential of using SPD methods: Here, the text just talks about HPT in the last sentence. Should the SPD be HPT? Or HPT is part of SPD methods?
Line 112 and 113, after HPT was 5.7: What is unit of 5.7 and definition of total deformation degree?
Line 118-120: This sentence contains two verbs.
Line 135: What does the 1/8 disk mean?
In Fig 2c: The label of 1, 2 and 3 is too small to see. Either change the color or size will do.
There are many places in the sentences, a comma should be put to make the sentence more readable. For example, line 81, line 117, line 171, line 190.
MMSC already defined in line 102, should not be defined again at line 159.
Author Response

(The authors gave the same response as above.)

Reviewer 3 Report
Manuscript ID: materials-2059047
Title: Structure, biodegradation and bioactivity in vitro of Zn-1%Mg alloy strengthened by high pressure torsion
The manuscript ID: materials-2059047 ”Structure, biodegradation and bioactivity in vitro of Zn-1%Mg alloy strengthened by high pressure torsion” studied the effect of high pressure torsion (HTP) on the microstructure, phase composition, mechanical characteristics, degradation rate, and bioactive properties of Zn-1% Mg alloy.
Must be admitted that the article does not introduce highly innovative aspects but it broadens the knowledge in the discussed scope.
The manuscript is well prepared with correct methods and good discussion of the results. The introduction provide sufficient background and include the most relevant references. The conclusions are clear and well supported by the results.
Author Response

(The authors gave the same response as above.)
